# A Comparative Study on the Drivers of Forest Fires in Different Countries in the Cross-Border Area between China, North Korea and Russia

Donghe Quan [1], Hechun Quan [1,2], Weihong Zhu [1,3], Zhehao Lin [1,2] and Ri Jin [1,2,3,*]

1   College of Geography and Ocean Sciences, Yanbian University, Hunchun 133300, China
2   Northeast Asian Research Center of Transboundary Disaster Risk and Ecological Security, Yanbian University, Hunchun 133300, China
3   Jilin Provincial Joint Key Laboratory of Changbai Mountain Wetland & Ecology, Changchun 130102, China
*   Correspondence: jinri0322@ybu.edu.cn; Tel.: +86-130-0908-3971

**Abstract:** The occurrence and spread of forest fires are the result of the interaction of many factors. In cross-border areas, different fire management systems may lead to different forest fire driving factors. A comparative analysis of the forest fire driving factors in different countries can help to provide ideas for fire prevention and control. In this study, based on the logistic regression (LR) model and standardized coefficients, we determined the relative impact of forest fire driving factors in different countries, in the cross-border area between China, North Korea and Russia from 2001 to 2020, and established a forest fire probability and fire risk level division using a Kriging interpolation. The results show that the climate is the most important factor affecting the probability of forest fires in the cross-border area, followed by the topography and vegetation factors; human activities have the least influence. From a country-by-country perspective, the forest fires on the Chinese side were more affected by humans than on the North Korean and Russian sides and they were mainly concentrated in areas with a low altitude and high population density. The forest fires on the North Korean side and the Russian side were more affected by nature than on the Chinese side and were mainly concentrated in areas with a low altitude, high temperature and little rainfall. The high-risk areas for forest fires were mostly concentrated near the border between China, North Korea and Russia, where transboundary fires pose a great threat to forest resources and rare animals. This study shows that there is a significant difference between the impact of different forest fire management systems on fire conditions, and that active forest fire control policies can effectively reduce the damage caused by forest fires. Due to the complexity of the geopolitics in cross-border areas, transboundary firefighting faces certain difficulties. In the future, it will be necessary to strengthen the cooperation between countries and establish transboundary joint defenses against forest fires in order to protect the declining forest resources and the habitats of rare animals.

**Keywords:** forest fire occurrence drivers; logistic regression; forest fire prediction model; fire risk zones; cross-border area

## 1. Introduction

Fire is an important ecological factor in forest ecosystems and appropriate forest fire has a positive impact on the regeneration and succession of tree species [1]; however, large-scale forest fires can destroy the forest structure and environment [2,3], damage wildlife and their habitats [4,5], reduce the biodiversity of forest ecosystems [6], and threaten the safety of human life and property [7,8]. In recent years, factors such as an increase in extreme climate conditions and population growth have led to an increase in the incidence of forest fires in different parts of the world to varying degrees [9,10], and the threat of fire to forest ecosystems has become more serious [11,12]. Therefore, exploring the driving

factors of forest fires and establishing effective forest fire prediction models can provide beneficial assistance for forest fire management.

Previous studies have shown that the occurrence and spread of forest fires is a result of the interaction of many factors [13,14], which can be roughly divided into the following categories: climate, vegetation, topography and human activities [15–18]. In high-temperature, dry and windy weather, the moisture content of combustibles is low, and their own temperature is high, so that little energy is required to reach an ignition point, which greatly increases the risk of forest fires [19]. Therefore, various weather indicators are often used to infer the risk index of forest fires, such as Canada's forest weather index (FWI) and Australia's forest fire danger index (FFDI), which are widely used around the world to measure the risk of forest fires [20,21]. The topography affects the local climate and the composition and spatial distribution of combustibles, resulting in different fire environments [22], while combustible vegetation provides the material basis for the occurrence of forest fire [23]. The state and load of the combustibles directly affect whether forest fires can occur and the speed of spread after they occur [24,25]. Moreover, human activities may cause fires to ignite. In most parts of the world, forest fires are mainly caused by human-made sources of ignition [26]. At the same time, human activity plays a major role in firefighting; for example, man-made roads may hinder the spread of fire and act as channels for firefighting [27].

In order to increase the adaptability of forest ecosystems and humans against forest fires, many researchers have conducted studies on the driving factors of forest fires around the world [15,18,23,28,29]. However, most studies have been conducted independently within a single country or region, and there has been a lack of comparability among these studies due to inconsistencies in the data sources, analytical methods and study scales. The effects of forest fire drivers in different regions may be diametrically opposed, due to the complexity of forest ecosystems and the widespread spatial heterogeneity of various geographical elements [30]. Particularly in transboundary regions, different fire management systems lead to different fire paths. For example, due to the different fire control policies and residents' lifestyles in different countries, different fire regimes are in place where there are similar forest tree species and climatic conditions [31]. Therefore, it is crucial to use a unified data source and statistical method to compare the differences in the forest fire drivers in different countries in cross-border regions, as this can help us to understand the impact of different fire management regimes and how fire drivers affect the fire occurrence in different political and socio-economic contexts.

According to previous studies, wildfires are common natural disturbances in the Northeast Asia forest region [32,33]. Due to the particularity of the geopolitics in cross-border areas, once a forest fire occurs, it is difficult for countries to implement cross-border cooperation in terms of their fire management, and it is difficult to control the fire in a timely and effective manner, which makes the fire more likely to threaten the local forest resources and residents. Since the catastrophic forest fire in the Greater Khingan Mountains in China in 1987, which resulted in serious losses of natural resources and human security [34], China has strengthened the control of forest fires in various ways, including through the prevention of border fires. However, with the increase in forest fire incidents worldwide and the complexity that exists within the cross-border areas [9,10], it is more urgent for us to explore the driving mechanisms of forest fires in those cross-border areas, and to classify the forest fire risks in these areas, so as to provide assistance for the next steps of forest fire prevention and control.

Taking the above factors into consideration, we selected the cross-border area between China, North Korea and Russia as the study area, and established a forest fire probability prediction model, aiming to analyze the relative impact of forest fire drivers on different countries in order to facilitate the development of forest fire risk-assessment and management strategies in the region. The specific goals were as follows: (1) to select different climatic conditions, terrains, vegetation and human activity driving-factors based on the logistic regression (LR) model to establish a forest fire probability prediction model, and to

explore the applicability of the model in Northeast Asia; (2) to analyze and compare the relative influence of different national forest fire driving-factors on forest fire occurrence; (3) to establish the forest fire occurrence probability and fire risk level in the cross-border areas, and provide suggestions for forest fire management in these areas.

## 2. Materials and Methods

### 2.1. Study Area

The cross-border areas between China, North Korea and Russia include northeastern China, northern North Korea and the Russian Far East, at 40°01′ N–51°26′ N, 125°16′ E–140°41′ E, with a total area of 540,000 square kilometers (Figure 1). The central part of this region contains the Sanjiang Plain and the Muling–Xingkai Plain, with a lower terrain; the Sikhote-Alin Mountains are on the eastern side and the Changbai Mountains are on the southern side, with a higher terrain. The whole region is in a temperate monsoon climate zone, with the same periods of rain and heat. The eastern water vapor from the Sea of Japan is blocked by the mountains; therefore, the precipitation shows longitudinal zonal differences on the eastern and western sides, and the temperature shows latitudinal zonal differences on the northern and southern sides (Table 1). The population is unevenly distributed, and is mainly concentrated on the Chinese and North Korean sides (Table 1).

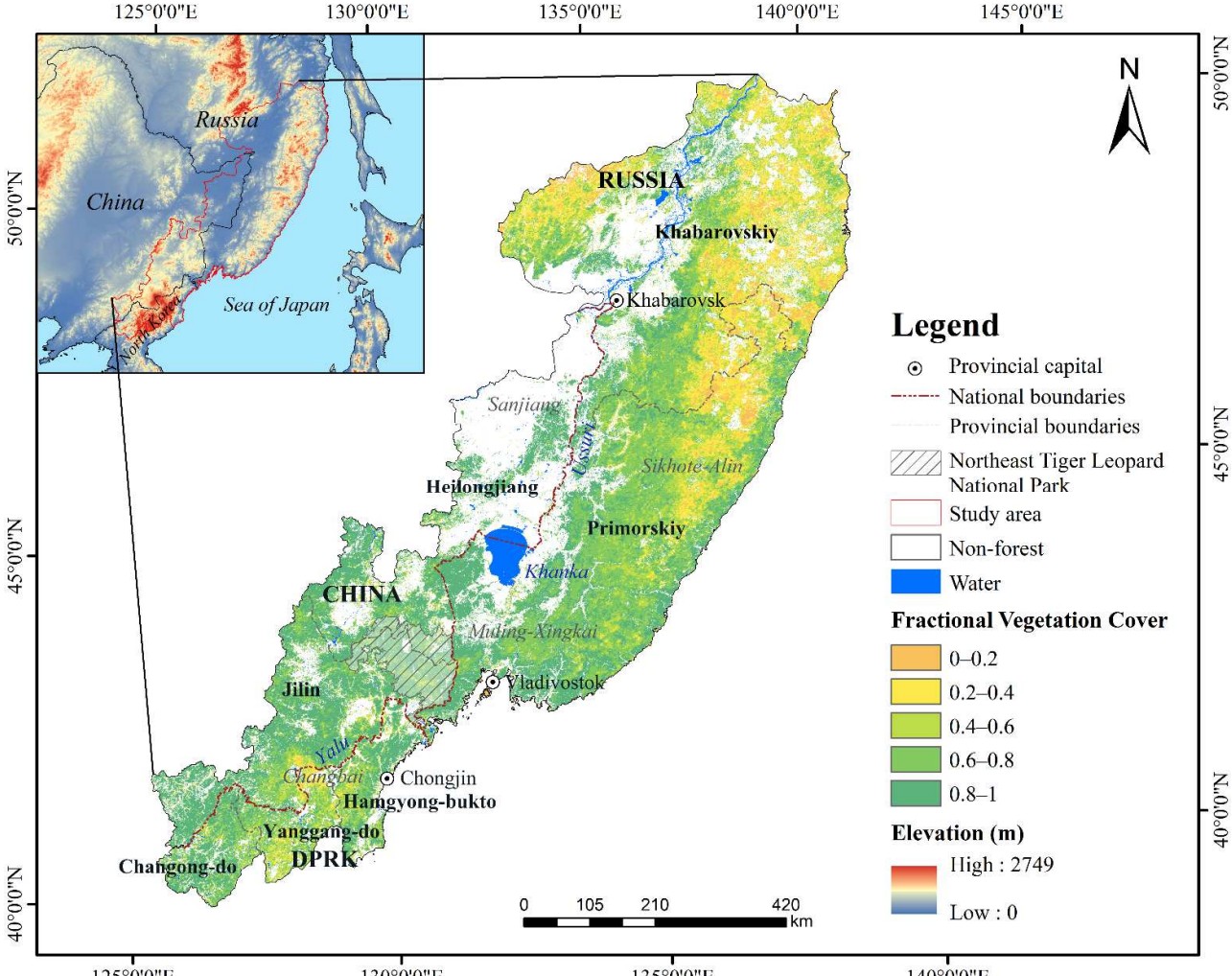

**Figure 1.** Study area. The white part in the study area represents no forest coverage. The calculation method of the fractional vegetation cover and other data sources are provided in Section 2.3.

The study area is rich in forest resources, with a forest coverage rate of nearly 70%, of mostly coniferous forest and mixed coniferous–broadleaved forest. The main tree species include larch (*Larix gmelinii* Rupr. Kuzen.), Korean pine (*Pinus koraiensis* Sieb. et Zucc.), fish scale spruce (*Picea jezoensis* var. *microsperma*), birch (*Betula platyphylla* Suk.) and Mongolian oak (*Quercus mongolica* Fisch. ex Ledeb.). The area is also the habitat of many rare animals and endangered animals such as the Amur tiger (*Panthera tigris* ssp. *altaica*) and Amur leopard (*Panthera pardus orientalis*). In recent years, with climate change and intensified human activities, forest fires have occurred frequently in the region [35]. In terms of the spatial distribution of forest fires, there are obvious differences between the different countries. According to the MOD14A1 monitoring results, the fire rate on the Russian side is much higher than that in China and North Korea. This is partly due to Russia's more passive forest fire management [36], and partly due to the fact that Russia is so sparsely populated that many fires occur in unpopulated areas [33].

**Table 1.** Basic conditions of different countries in the study area.

| Area | Natural Conditions | Human Conditions and Forest Fire Management System | Fire Regime |
|---|---|---|---|
| Chinese side | The total area is 150,000 square kilometers, and the forest coverage rate is 55%. The terrain is low in the north and high in the south, with an average elevation of 450 m. The area has a temperate humid and semi-humid continental monsoon climate. The average temperature in January is between −21 °C and −18 °C, and the average temperature in July is between 21 °C and 22 °C. The annual precipitation is 500–650 mm. | The total population is approximately 4.7 million. When fighting large-scale forest fires, China generally uses firefighting aircraft to create artificial rain or to spray chemical agents, and cooperate with ground-based forest firefighting forces to carry out ground–air integrated firefighting [37]. For high-risk areas of fire, manpower is deployed in advance, and protective forest belts are built to prevent the spread of wildfires. | According to the monitoring results of MOD14A1, a total of 4001 forest fires occurred in the fire season (March–November) from 2001 to 2020. |
| North Korean side | The total area is 48,500 square kilometers, and the forest coverage rate is 78%. The overall terrain is high, with an average elevation of 920 m. The area has a temperate monsoon climate. The annual average temperature is between 2 °C and 5 °C. The annual precipitation is 650–700 mm. | The total population is approximately 4.3 million. To date, most of North Korea's firefighting has been with manpower. Under the mobilization of the government, the general public actively participate in firefighting; however, due to the lack of modern firefighting facilities, this often requires significant manpower [38]. | According to the monitoring results of MOD14A1, a total of 8143 forest fires occurred in the fire season (March–November) from 2001 to 2020. |
| Russian side | The total area is 348,000 square kilometers, and the forest coverage rate is 72%. The terrain is high in the east and low in the west, with an average elevation of 420 m. The area has a temperate oceanic monsoon climate. The average temperature in January is between −30 °C and −12 °C, and the average temperature in July is between 14 °C and 21 °C. The annual precipitation is 600–900 mm. | The total population is approximately 2.5 million. Russia abolished the national forest protection system after introducing a new Forest Code in 2007, and also takes a negative attitude towards wildfires in forest areas where people are scarce and difficult to reach [36]. | According to the monitoring results of MOD14A1, a total of 87,543 forest fires occurred in the fire season (March–November) from 2001 to 2020. |

### 2.2. Overarching Study Design

First, we used satellite monitoring data and land use data to extract fire points in the forest area within the study area, and we used ArcGIS 10.8 to randomly create a corresponding number of non-fire points in the unburned area, extract the climatic, topographic, vegetation and human activity factors at the corresponding time and place, and establish a forest fire and fire environment database in the study area (all data sources are detailed

in Section 2.3.). Secondly, LR was performed on the three different countries in the study area to establish the forest fire probability prediction model, and to ensure that the sample size and data sources of the different countries participating in the model fitting were consistent. Thirdly, according to the partially standardized logistic regression coefficients, we compared and analyzed the relative importance of the driving factors of forest fires in the different countries. Finally, according to the prediction value of the forest fire probability prediction model, we constructed the spatial distribution of the forest fire occurrence probability and the division of the fire danger level, and we put forward the key points of forest fire prevention and control for the future. The overall technology roadmap is shown in Figure 2.

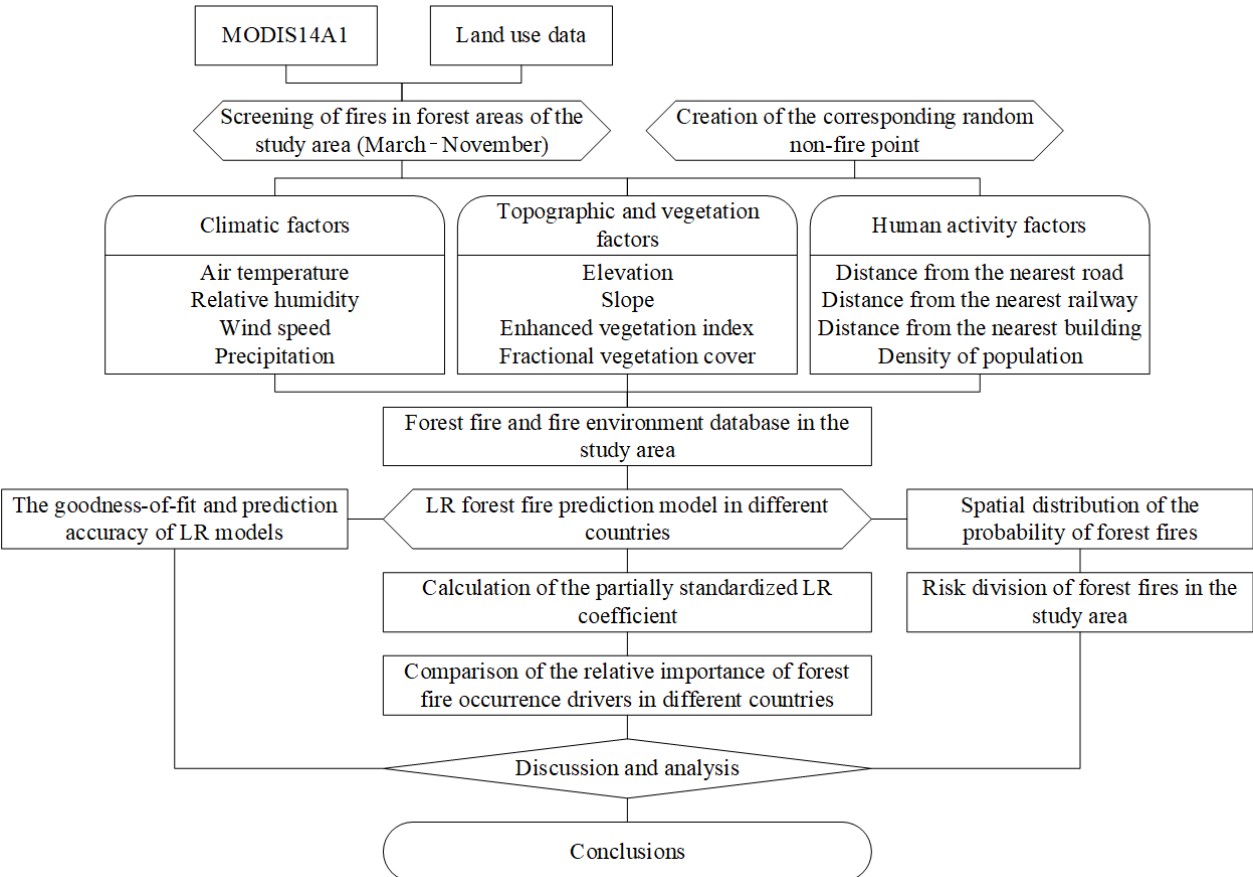

**Figure 2.** The overall technology roadmap of this study.

### 2.3. Data Source and Processing

2.3.1. Fire Point Data Extraction and the Creation of Random Points

The fire point data were based on the 1 km resolution MODIS vegetation fire data product, MOD14A1, from 2001 to 2020 (https://ladsweb.nascom.nasa.gov/, accessed on 22 September 2022), combined with the 30 m resolution land use data provided by the GlobeLand30: Global Geo-information Public Product (http://www.globallandcover.com/, accessed on 22 September 2022), to extract the fire point information for the fire season (March–November) within the forest range of the study area. A total of 99,687 fire points were extracted, including 4001 on the Chinese side, 8143 on the North Korean side, and 87,543 on the Russian side.

The dependent variable required for the modeling of the LR was a binary variable. Since this study was a comparative study of the driving factors of forest fires in different countries, 4000 fire points were taken as the total fire point sample from each country randomly. Using ArcGIS 10.8, we randomly selected points within the study area as non-

fire points. Since the resolution of the MODIS vegetation fire data is 1 km, we used the land use data and a buffer 1 km away from a fire point to remove the non-fire points outside of the forest range and less than 1 km away from the fire point [17]. We randomly selected 6000 non-fire points that were 1.5 times the size of the fire point in each country as the total non-fire point samples [15], and then used the "RANDBETWEEN" function of Microsoft Excel 2019 to assign random dates within the fire season to the non-fire points, to ensure a randomness in both time and space.

### 2.3.2. Driving Factors

In this study, a total of 12 variables, including climate, topography, vegetation, and human activities, were selected as the driving factors affecting the occurrence of forest fires. The specific description of each variable and the data source are shown in Table 2.

**Table 2.** Overview of independent variables used in the construction of forest fire prediction models.

| Variable Type | Variable Name | Code | Resolution /Scale | Source |
|---|---|---|---|---|
| Climatic | Mean daily air temperature at sigma level 995 | Temp | 2.5°/°C | NCEP-NCAR Reanalysis 1 data and CPC Global Unified Gauge-Based Analysis of Daily Precipitation data were provided by the NOAA Physical Sciences Laboratory, Boulder, CO, USA (https://psl.noaa.gov/, accessed on 22 September 2022). |
| | Mean daily relative humidity at sigma level 995 | Rhum | 2.5°/% | |
| | Mean daily wind velocity at sigma level 995 | Wind | 2.5°/m/s | |
| | Daily total of precipitation | Pre | 0.5°/mm | |
| Topographic | Elevation | Elev | 30 m/m | ASTER GDEM was provided by the Geospatial Data Cloud site, Computer Network Information Center, Chinese Academy of Sciences (http://www.gscloud.cn/, accessed on 22 September 2022). |
| | Slope | Slope | 30 m/° | |
| Vegetation | Monthly Enhanced Vegetation Index | EVI | 1 km | MOD13A3—MODIS/Terra Vegetation Indices Monthly L3 Global 1 km SIN Grid dataset was acquired from the Level-1 and Atmosphere Archive & Distribution System (LAADS) Distributed Active Archive Center (DAAC), located in the Goddard Space Flight Center in Greenbelt, Maryland (https://ladsweb.nascom.nasa.gov/, accessed on 22 September 2022). |
| | Annual Fractional Vegetation Cover | FVC | 1 km/% | |
| Human activity | Distance from the nearest road | Dis_road | km | Global road and building dataset was provided by the OpenStreetMap Foundation (http://download.geofabrik.de/, accessed on 22 September 2022). |
| | Distance from the nearest railway | Dis_railway | km | |
| | Distance from the nearest building | Dis_building | km | |
| | Density of population | POP | 100 m/number | Global population dataset was provided by the WorldPOP Hub (https://hub.worldpop.org/, accessed on 22 September 2022). |

#### Climatic Factors

The climatic factors include the daily average temperature, daily average relative humidity, daily average wind speed, and daily cumulative precipitation, and the data were derived from the gridded climatic dataset provided by the National Oceanic and Atmospheric Administration Physical Sciences Laboratory (Table 2). The daily average temperature, daily average relative humidity, and daily average wind speed were derived from the 2.5° × 2.5° resolution NCEP/NCAR Reanalysis 1 dataset [39], and the daily cumulative precipitation was derived from the 0.5° × 0.5° resolution CPC Global Unified Gauge-Based Analysis dataset [40].

MATLAB R2021a was used to read each gridded meteorological dataset, and extract the daily values corresponding to all the sample points in the dataset according to the

spatial and temporal information of the sample points. The daily average wind speed was calculated based on the daily average U-wind and daily average V-wind.

Topographic Factors

Topographic factors include the elevation and slope. The elevation data were taken from the ASTER GDEM 30 M resolution digital elevation data provided by the Geospatial Data Cloud (Table 2), using the 3D analysis tool of ArcGIS 10.8 to generate the slope raster data, and then using the extraction tools to extract the elevation and slope corresponding to the sample points.

Vegetation Factors

In this study, the enhanced vegetation index (EVI) and fractional vegetation cover (FVC) were selected as the initial vegetation variables affecting the occurrence of forest fires. EVI is an important index used to characterize forest vegetation coverage, which addresses the problem of an easy saturation of the normalized difference vegetation index (NDVI), and effectively reduces atmospheric and soil background noise [41]. The FVC is the percentage of the vertical projected area of the canopy or branch area of the vegetation in the unit area, and is used to characterize the total amount of live and dead combustibles above the surface [42]. We estimated the FVC based on the EVI using a pixel dichotomy model [43]; the formula is as follows:

$$FVC = \frac{EVI - EVIsoil}{EVIveg - EVIsoil} \tag{1}$$

where $EVI_{soil}$ is the EVI value of pure soil pixels and $EVI_{veg}$ is the EVI value of pure vegetation pixels. Affected by the vegetation growth conditions and vegetation types, the EVI values of remote sensing images in different periods will vary over time and space; therefore, it is not advisable to use uniform $EVI_{soil}$ and $EVI_{veg}$ in the pixel dichotomous model [42]. Usually, the minimum and maximum values within the confidence interval are used to assign values [44]. In this study, a 2%–98% confidence interval was used to estimate the FVC in this area.

The EVI data were taken from the MOD13A3 product, which provides a monthly sinusoidal projection grid product with a resolution of 1 km (Table 2). Using the MODIS Reprojection Tool (MRT) to extract and reproject the EVI bands in MOD13A3, a monthly EVI raster map was obtained. Using the maximum value composite tool in ENVI 5.3, the maximum EVI value of each year was obtained. We used the compute statistics tool to determine the confidence interval and obtain the values of the $EVI_{soil}$ and $EVI_{veg}$; then, we used the band math tool to calculate the interannual FVC. Finally, we used ArcGIS 10.8 to extract the monthly EVI and annual FVC corresponding to the sample points.

Human Activity Factors

Human activity factors include the distance from the nearest railway, distance from the nearest road, distance from the nearest buildings and the population density. Among these, the railway, road and building vector data were obtained from Open Street Map (Table 2). The population data were obtained from the annual population counts provided by World POP (Table 2), with a precision of 100 m.

Using the ArcGIS 10.8 proximity toolset, the distance from a railway, distance from a road and distance from buildings were calculated for each sample point, and the extraction tools were used to extract the population density corresponding to the sample point.

*2.4. Methodology*

2.4.1. Logistic Regression Model

LR models are commonly used to describe the relationship between a binary dependent variable (0 or 1) and one or more independent variables. As a forest fire probability prediction model, LR has been widely used internationally to analyze the driving factors

of forest fires [15–17,28,29]. In this study, the probability of the occurrence of a forest fire (y = 1) was set as P, and the probability of no occurrence of a forest fire (y = 0) was set as (1 − P); then, the LR between the probability of the occurrence of a forest fire and the respective variables could be established:

$$\text{logit}(P) = \ln\left(\frac{P}{1 - P}\right) = \alpha_0 + \alpha_1 x_1 + \alpha_2 x_2 + \ldots + \alpha_n x_n \tag{2}$$

where P is the probability of an occurrence of a forest fire, $(x_1, x_2, \ldots, x_n)$ denotes the independent variable that affects the occurrence of a forest fire, n is the number of independent variables and $(\alpha_1, \alpha_2, \ldots, \alpha_n)$ is the LR correlation coefficient of each independent variable.

2.4.2. Model Variable Selection

Multicollinearity refers to the precise correlation or a high correlation between different independent variables in the linear regression model, which leads to a distortion of the model prediction and a loss in significance of the variable significance tests [45]. In this study, the variance inflation factor (VIF) was used to test the independent variables in the model to exclude those variables with a significant collinearity. Most studies assume that when VIF > 10, there is significant collinearity between the independent variables [46], and the variables with a significant collinearity were excluded based on this standard.

In this study, 80% of the total sample dataset of the three countries (i.e., 10,000 sample points in each country) were used as the modeling samples of the country, and 20% were used as the independent test samples. A total of 70% of the modeling samples of each country were used as the training samples, and 30% were used as the validation samples. In order to reduce the influence of the randomness in the sample partitioning on the model parameter selection, this process was repeated five times. A total of 15 subsample datasets from the 3 countries were obtained, and each subsample dataset included a training sample dataset and a validation sample dataset. The LR model was fitted to each training sample of the three countries by a stepwise regression, and the validation samples were used for verification. The variables with at least 3 significant correlations ($\alpha < 0.05$) were selected from the 5 intermediate models in each country to perform an optimal model fitting on their modeling data. Finally, the optimal model was independently verified using independent test samples from each country.

2.4.3. Prediction Accuracy of the Models

In this study, the value of the area under the curve (AUC) of the receiver operating characteristic (ROC) curve was used to verify the accuracy of the model fitting. In recent years, many studies at home and abroad have used this method to judge the goodness-of-fit of the binary classification model [47]. Typically, AUC values range from 0.5 to 1.0, with a larger AUC representing a better performance, and an AUC > 0.8 indicating that the model has a good predictive ability.

Many previous studies have used the system default of 0.5 as the cut-off value for the binary classification model, so as to classify the predicted probability of the model [48]. However, an increasing number of studies have suggested that, in a binary model with an uneven number of positive and negative samples, such as in a forest fire probability prediction model, the sample size without fire must be larger than the sample size with fire to conform to the reality, and using a default value of 0.5 will greatly reduce the prediction accuracy of fire events [49]. In order to solve this problem, in recent years, most scholars have calculated the Youden index (i.e., sensitivity coefficient + specificity coefficient − 1) based on the sensitivity and specificity index in the ROC curve to determine the optimal cut-off value of the LR model [50]. If the predicted value of the model is higher than the cut-off value, it is judged that a forest fire will occur, and if it is lower than the cut-off value, it is judged that no forest fire will occur. At the same time, the prediction accuracy of each training sample, validation sample, modeling sample and independent test sample dataset is calculated according to the actual value of each sample point.

### 2.4.4. Evaluating the Relative Importance of the Driving Factors

The standardized regression coefficient refers to the regression coefficient obtained after standardizing the independent variable and dependent variable at the same time. It is used to compare the effect of different independent variables on the dependent variable after eliminating the influence of differences in the dimension and magnitude of the data. We used partially-standardized logistic regression coefficients to assess the relative importance of each variable to the forest fire occurrence. The greater the absolute value of the partially-standardized logistic regression coefficient, the greater the change in the predicted probability of a forest fire when the independent variable changes by one standard deviation and the other independent variables remain unchanged [51]. The calculation method of the partially-standardized logistic regression coefficient proposed by Menard was used [52]:

$$b^*_{SAS} = (b)(s_X)/(\pi/\sqrt{3}) = (b)(s_X)/1.8138 \tag{3}$$

where b is the unstandardized logistic regression coefficient and $s_X$ is the sample standard deviation of the independent variable X.

### 2.4.5. Generation of a Forest Fire Occurrence Probability Map and Fire Risk Classification

The LR model was used to calculate the fire occurrence probability of the fire points and the non-fire points in the study area, the Kriging interpolation method was used to interpolate the spatial distribution of the forest fire occurrence probability in the study area, and a spatial distribution map of the forest fire occurrence probability was obtained. According to the current status of domestic and international research, the fire risk level of the study area was divided by the optimal cut-off value of the LR model and 0.5 [15]. The area with the probability of a forest fire occurrence less than the cut-off value was classified as a low-fire-risk area, the area with the probability of a forest fire occurrence between the cut-off value and 0.5 was classified as a medium-fire-risk area, and the area with a forest fire probability greater than 0.5 was classified as a high-fire-risk area, demonstrating the forest fire risk level division of the study area.

## 3. Results

### 3.1. Fitting Results of the Forest Fire Occurrence Probability Prediction Model

#### 3.1.1. Multicollinearity Test Results

After the multicollinearity test, the VIF values of all of the initial variables in the three countries were less than 10, and there was no multicollinearity among the variables; therefore, the next step of the model fitting could be performed. The multicollinearity test results are shown in Table 3.

**Table 3.** The results of multicollinearity diagnosis.

| Initial Variable | VIF | | |
|:---:|:---:|:---:|:---:|
| | China | North Korea | Russia |
| Temp | 2.265 | 1.929 | 1.827 |
| Rhum | 1.181 | 1.383 | 1.358 |
| Wind | 1.053 | 1.167 | 1.059 |
| Pre | 1.213 | 1.146 | 1.146 |
| Elev | 1.187 | 1.286 | 2.449 |
| Slope | 1.113 | 1.039 | 1.428 |
| EVI | 2.513 | 2.188 | 1.964 |
| FVC | 1.170 | 1.150 | 1.869 |
| Dis_road | 1.243 | 1.364 | 1.645 |
| Dis_railway | 1.224 | 1.108 | 1.776 |
| Dis_building | 1.375 | 1.309 | 2.471 |
| POP | 1.034 | 1.147 | 1.039 |

### 3.1.2. Model Parameter Fitting Results

The LR model fitting was performed using SPSS 25.0 software for the training samples in five subsamples for each country. Stepwise regression was used to remove the insignificant variables, and five intermediate models were obtained. Variables with three or more significant correlations ($\alpha < 0.05$) in the five intermediate models were chosen to enter the final fitting of the modeling samples. The fitting results are shown in Table 4.

**Table 4.** The fitting results of the final model parameters of LR in different countries.

| Variable | Country | Significant Correlation Times | Parameter Estimation | | | |
|---|---|---|---|---|---|---|
| | | | Coefficient | Standard Error | Wald Chi-Squared Value | Significance |
| Temp | China | 5 | 0.0795 | 0.0053 | 223.4483 | <0.0001 |
| | North Korea | 5 | 0.1338 | 0.0061 | 475.5498 | <0.0001 |
| | Russia | 5 | 0.0961 | 0.0052 | 342.3678 | <0.0001 |
| Rhum | China | 5 | −4.3544 | 0.2788 | 243.9326 | <0.0001 |
| | North Korea | 5 | −8.0400 | 0.2708 | 881.2861 | <0.0001 |
| | Russia | 5 | −7.2572 | 0.3468 | 438.0091 | <0.0001 |
| Wind | China | 0 | / | / | / | / |
| | North Korea | 5 | 0.0648 | 0.0145 | 19.9621 | <0.0001 |
| | Russia | 3 | 0.0328 | 0.0151 | 4.7469 | 0.0294 |
| Pre | China | 5 | −0.3008 | 0.0312 | 93.2182 | <0.0001 |
| | North Korea | 5 | −0.5385 | 0.0546 | 97.2522 | <0.0001 |
| | Russia | 5 | −1.0631 | 0.0890 | 142.7199 | <0.0001 |
| Elev | China | 5 | −0.0055 | 0.0002 | 1310.0693 | <0.0001 |
| | North Korea | 5 | −0.0014 | <0.0001 | 308.1419 | <0.0001 |
| | Russia | 5 | −0.0039 | 0.0002 | 446.3535 | <0.0001 |
| Slope | China | 5 | −0.0501 | 0.0049 | 105.7375 | <0.0001 |
| | North Korea | 0 | / | / | / | / |
| | Russia | 1 | / | / | / | / |
| EVI | China | 5 | −0.0005 | <0.0001 | 384.4244 | <0.0001 |
| | North Korea | 5 | −0.0008 | <0.0001 | 727.9396 | <0.0001 |
| | Russia | 5 | −0.0007 | <0.0001 | 537.2400 | <0.0001 |
| FVC | China | 3 | −0.5526 | 0.2285 | 5.8515 | 0.0156 |
| | North Korea | 3 | 0.4519 | 0.1873 | 5.8202 | 0.0158 |
| | Russia | 5 | −0.7643 | 0.1749 | 19.1006 | <0.0001 |
| Dis_road | China | 0 | / | / | / | / |
| | North Korea | 4 | −0.0996 | 0.0227 | 19.2222 | <0.0001 |
| | Russia | 4 | 0.0229 | 0.0063 | 13.2112 | 0.0003 |
| Dis_railway | China | 0 | / | / | / | / |
| | North Korea | 0 | / | / | / | / |
| | Russia | 5 | −0.0063 | 0.0008 | 64.2092 | <0.0001 |
| Dis_building | China | 4 | −0.0098 | 0.0034 | 8.5992 | 0.0034 |
| | North Korea | 3 | 0.0398 | 0.0122 | 10.6846 | 0.0011 |
| | Russia | 3 | −0.0082 | 0.0038 | 4.5439 | 0.0330 |
| POP | China | 5 | 0.2011 | 0.0495 | 16.5201 | <0.0001 |
| | North Korea | 5 | −0.4052 | 0.0783 | 26.7928 | <0.0001 |
| | Russia | 2 | / | / | / | / |
| Constant | China | / | 7.4033 | 0.2974 | 619.5687 | <0.0001 |
| | North Korea | / | 6.8816 | 0.2683 | 657.8327 | <0.0001 |
| | Russia | / | 8.9320 | 0.3580 | 622.3646 | <0.0001 |

Note: The number of significant correlation times refers to the number of times the variable is significantly correlated across the five intermediate models. Variables with significant correlation times less than 3 did not enter the final model fit; therefore, there is no parameter estimate for the final model.

The results show that, with the exception of the three variables of the daily average wind speed, distance from a road, and distance from a railway on the Chinese side, all the other variables entered the final model fitting. The daily average temperature and population density were positively correlated with the probability of a forest fire; the daily average relative humidity, daily cumulative precipitation, elevation, slope, EVI, FVC and distance from buildings were negatively correlated with the probability of a forest fire. With the exception of the two variables of the slope and distance from a railway on the North Korean side, all the other variables entered the final model fitting. The daily average temperature, daily average wind speed, FVC and distance from buildings were positively correlated with the probability of a forest fire; the daily average relative humidity, daily cumulative precipitation, elevation, EVI, distance from a highway, and population density were negatively correlated with the probability of a forest fire. With the exception of the two variables of the slope and population density on the Russian side, all the other variables entered the final model fitting. The daily average temperature, daily average wind speed and distance from a highway were positively correlated with the probability of a forest fire; the daily average relative humidity, daily cumulative precipitation, elevation, EVI, FVC, distance from a railway and distance from buildings were negatively correlated with the probability of a forest fire.

### 3.1.3. Model Prediction Accuracy Results

The sensitivity and specificity indices of each intermediate model and the final model were obtained using the ROC curve, and the optimal cut-off value for each model was calculated. The goodness-of-fit for each model was evaluated according to the AUC, and the prediction accuracy of each intermediate model was calculated using the validation samples in each subsample, as shown in Table 5. The AUC values of each intermediate model on the Chinese side were between 0.9136 and 0.9166, the model prediction accuracy was between 83.44% and 85.79%, and the AUC value of the final model was 0.9132. The AUC values of the intermediate models on the North Korean side were between 0.8956 and 0.9006, the model prediction accuracy was between 79.88% and 82.08%, and the AUC value of the final model was 0.8973. The AUC values of the intermediate models on the Russian side were between 0.9092 and 0.9217, the model prediction accuracy was between 83.85% and 85.79%, and the AUC value of the final model was 0.9153. The independent test results of the final model of the independent test samples from the different countries are shown in Table 6.

**Table 5.** Comparison of goodness-of-fit and prediction accuracy of LR models in different countries.

| Sample | Country | Cut-Off | AUC Value | Prediction Accuracy (%) | |
| --- | --- | --- | --- | --- | --- |
| | | | | Training | Validation |
| Sample 1 | China | 0.4885 | 0.9141 | 85.79 | 84.81 |
| | North Korea | 0.4116 | 0.9003 | 82.08 | 80.19 |
| | Russia | 0.4567 | 0.9092 | 83.85 | 85.59 |
| Sample 2 | China | 0.4414 | 0.9155 | 85.44 | 84.06 |
| | North Korea | 0.3675 | 0.9001 | 81.31 | 79.88 |
| | Russia | 0.4726 | 0.9217 | 85.79 | 83.91 |
| Sample 3 | China | 0.3734 | 0.9166 | 84.48 | 83.44 |
| | North Korea | 0.4030 | 0.8956 | 81.21 | 81.59 |
| | Russia | 0.4835 | 0.9106 | 84.71 | 85.47 |

**Table 5.** *Cont.*

| Sample | Country | Cut-Off | AUC Value | Prediction Accuracy (%) | |
|---|---|---|---|---|---|
| | | | | **Training** | **Validation** |
| | China | 0.4045 | 0.9163 | 84.88 | 83.91 |
| Sample 4 | North Korea | 0.3911 | 0.9006 | 81.63 | 81.59 |
| | Russia | 0.5276 | 0.9183 | 85.69 | 84.84 |
| | China | 0.4267 | 0.9136 | 85.21 | 84.19 |
| Sample 5 | North Korea | 0.3909 | 0.9006 | 81.63 | 80.50 |
| | Russia | 0.4098 | 0.9121 | 83.85 | 84.97 |
| Modeling | China | 0.4273 | 0.9132 | | |
| sample | North Korea | 0.3834 | 0.8973 | | |
| | Russia | 0.4867 | 0.9153 | | |

Note: For the accuracy test of the modeling samples, see Table 6.

**Table 6.** Prediction accuracy results for modeling samples and independent testing samples.

| Observed | | Predicted | | | | | |
|---|---|---|---|---|---|---|---|
| | | Modeling | | | Independent Test | | |
| | | **Non-Fire** | **Fire** | **Correct Rate** | **Non-Fire** | **Fire** | **Correct Rate** |
| | Non-fire | 4052 | 738 | 84.59 | 1025 | 185 | 84.71 |
| China | Fire | 480 | 2730 | 85.05 | 96 | 694 | 87.85 |
| | Overall pct. | | | 84.78 | | | 85.95 |
| | Non-fire | 3792 | 1011 | 78.95 | 936 | 261 | 78.20 |
| North Korea | Fire | 491 | 2706 | 84.64 | 117 | 686 | 85.43 |
| | Overall pct. | | | 81.23 | | | 81.10 |
| | Non-fire | 4197 | 617 | 87.18 | 1051 | 135 | 88.62 |
| Russia | Fire | 577 | 2609 | 81.89 | 160 | 654 | 80.34 |
| | Overall pct. | | | 85.08 | | | 85.25 |

The results show that the AUC values and model prediction accuracy of each intermediate model and the final model were high and similar, indicating that the overall fitting degree of each model was good. The overall prediction accuracy for the North Korean side (81.10%–81.23%) was lower than that of the Chinese side and the Russian side, but the prediction accuracy of the fire point on the North Korean side (84.64%–85.43%) was higher than that on the Russian side (80.34%–81.89%). On the Chinese side, the prediction accuracy of the fire point (85.05%–87.85%) and the overall prediction accuracy (84.78%–85.95%) were relatively high, and the prediction ability for forest fires was better than that of the North Korean side and the Russian side.

*3.2. Comparison of the Relative Importance of Forest Fire Occurrence Drivers in Different Countries*

Figure 3 shows the size of the partially standardized logistic regression coefficient for each variable in the LR model fitting process. On the whole (Figure 3a), climatic factors were the most important factors affecting the probability of forest fires, followed by the topography and vegetation factors, and human activity factors had the least influence.

In terms of the relative importance among the climatic factors (Figure 3b), the daily accumulated precipitation was the most important, and the influence on the Russian side was much larger than that on the North Korean side and the Chinese side ($|-2.627| > |-1.857| > |-0.823|$, respectively). This was followed by the daily average temperature and daily average relative humidity; the relative importance of the two was basically the same, and the influence on the North Korean side was slightly larger than that on the Russian side and the Chinese side (Temp $|0.570| > |0.487| > |0.411|$, and Rhum $|-0.703| > |-0.476| > |-0.325|$, respectively). The average daily wind speed was not selected as

a significant variable in China, and had the lowest relative importance in the other two countries, where the influence on the North Korean side was slightly higher than that on the Russian side (|0.082| > |0.042|, respectively).

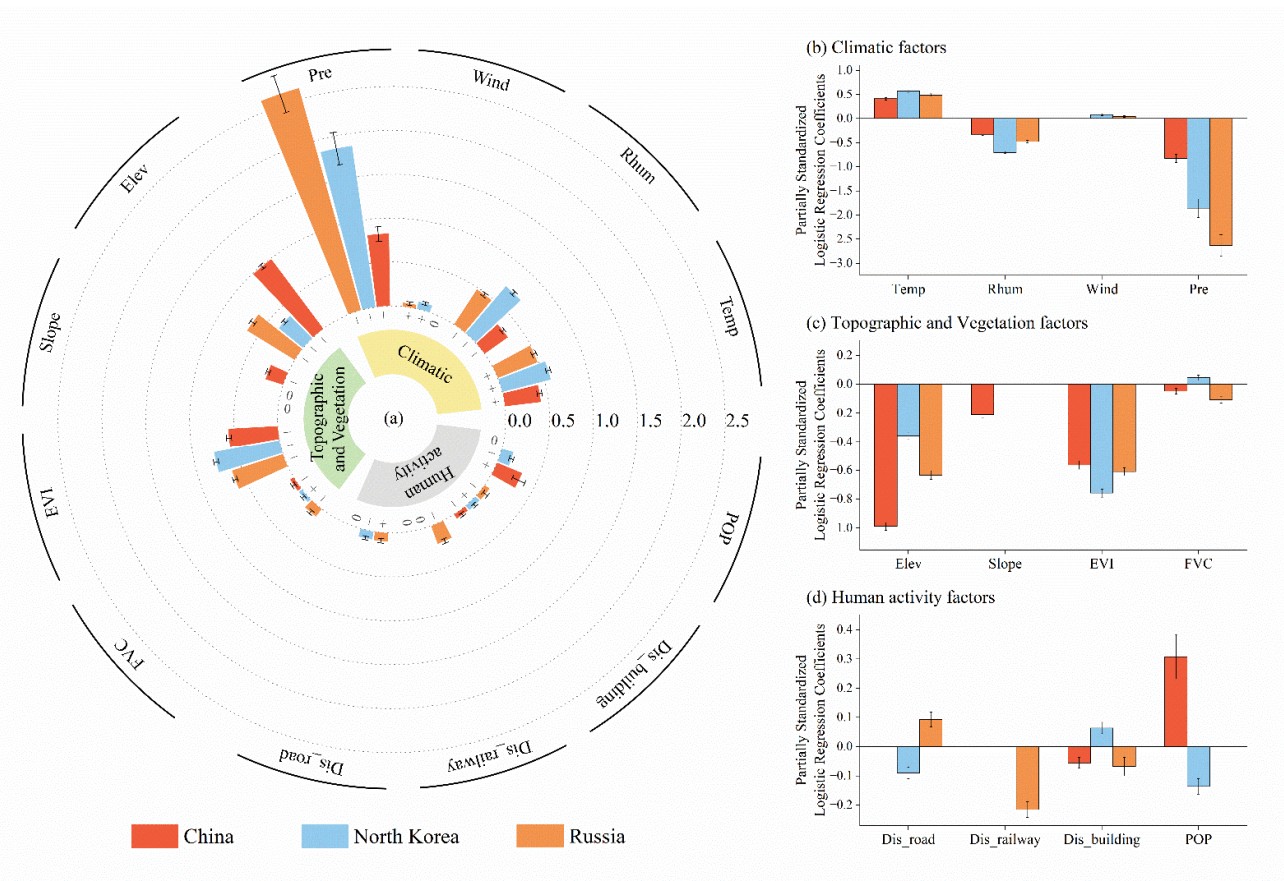

**Figure 3.** Comparison of the relative importance of forest fire occurrence drivers in different countries. (**a**) Comparison of partially standardized logistic regression coefficients for all drivers. The outermost arcs show different variable names, with "+" at the bottom of each column meaning that the coefficient was positive, "−" meaning that the coefficient was negative and "0" meaning that the coefficient did not pass the significance test in the previous stepwise regression. (**b**) Comparison of partially standardized logistic regression coefficient of climatic factors, (**c**) topographic and vegetation factors, and (**d**) human activity factors.

In terms of the relative importance between the topography and vegetation factors (Figure 3c), the relative importance of elevation and EVI was much greater than that of the slope and FVC. In China, the relative importance of the elevation (|−0.988|) was the highest, followed by the EVI (|−0.564|), slope (|−0.210|) and FVC (|−0.048|). In Korea, the slope was not selected as a significant variable; EVI (|−0.758|) had the highest relative importance, followed by the elevation (|−0.357|) and FVC (|0.045|). In Russia, the slope was not selected as a significant variable; the elevation (|−0.634|) had the highest relative importance, followed by the EVI (|−0.611|) and FVC (|−0.107|).

The relative importance of human activity factors varied greatly in the different countries (Figure 3d). In China, the distance from a road and distance from a railway were not selected as significant variables, and the impact of the POP (|0.308|) on the forest fire occurrence was much higher than that of the distance from buildings (|−0.055|). In North Korea, the distance from a railway was not selected as a significant variable, and the other three variables generally had a low impact on the forest fire occurrence, in a descending order of the POP (|−0.136|), the distance from a road (|−0.090|) and distance from buildings (|0.064|). In Russia, the POP was not selected as a significant variable;

the relative importance of the distance from a railway (|−0.215|) was relatively high, and the distance from a road (|0.093|) and the distance from buildings (|−0.068|) were relatively low.

### 3.3. Spatial Distribution of Forest Fire Probability and Fire Risk Division

Based on the predicted value of the forest fire probability model, the Kriging interpolation method was used to obtain the spatial distribution of the forest fire probability in the study area (Figure 4a). According to the optimal cut-off value of each model (i.e., 0.4273 in China, 0.3834 in North Korea and 0.4867 in Russia) and using 0.5 as the division criterion, the fire danger level of the study area was divided. Values less than the optimal cut-off value were the low-fire-risk areas, the optimal cut-off value and 0.5 were the medium-fire-risk areas, and those with a value higher than 0.5 were the high-fire-risk areas (Figure 4b).

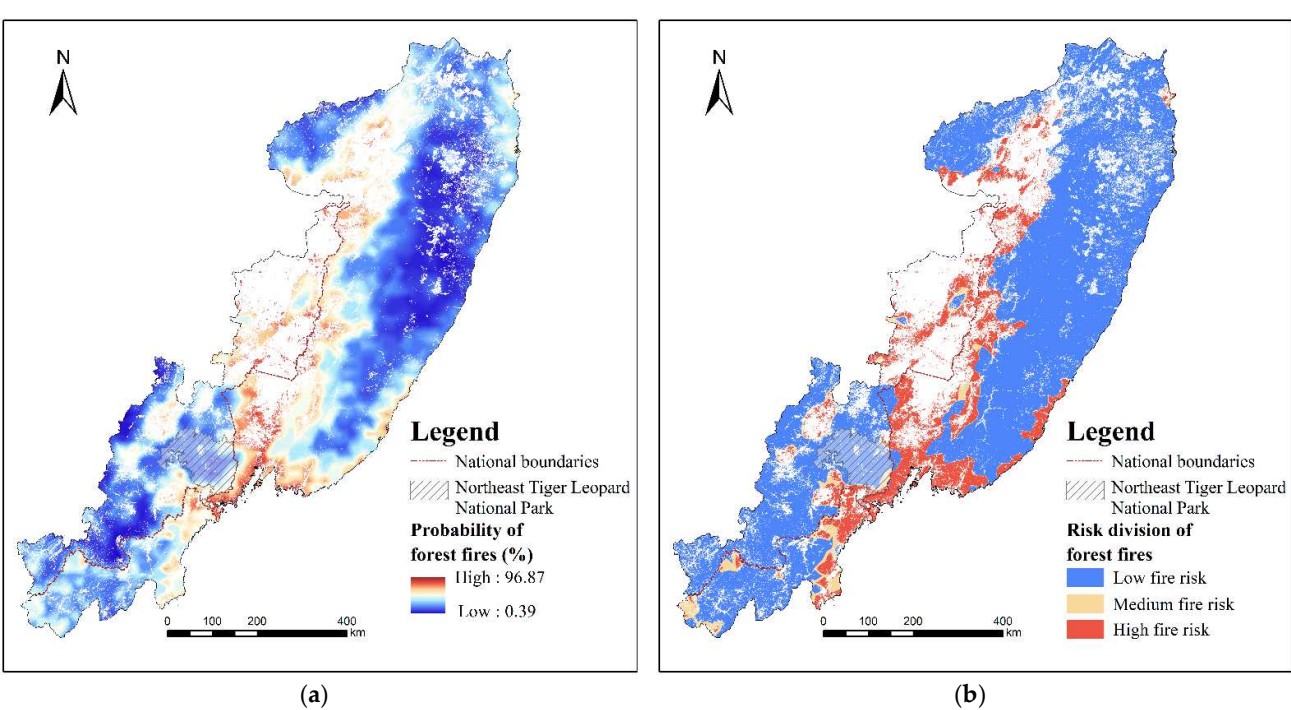

(**a**)                  (**b**)

**Figure 4.** (**a**) Spatial distribution map of forest fire probability in the study area. (**b**) Fire risk level zoning map of the study area.

## 4. Discussion

### 4.1. Applicability of the Logistic Regression Model in the Study Area

Judging from the AUC value and prediction accuracy, the LR model had a good prediction ability in the study area. In previous forest fire prediction studies using the LR model, the AUC value of the LR model was generally around 0.75–0.87, and the prediction accuracy was generally 65%–80% [15,16,28,29]. In contrast, the LR model in this study had a higher applicability. First, this may be due to the comprehensive selection of variables in this study, taking into account both natural and human factors. Second, from a statistical point of view, the accuracy of statistical models largely depends on the size of the sample and the accuracy of the data. The data selected in this study had a long-time span and a large sample size. In terms of the data accuracy, previous studies have mostly used monthly, quarterly or annual averages for their climatic data [15,17], but for forest fire regimes, the cause and spread of a fire mainly depend on the fire environment at the time of the fire; therefore, it is necessary to improve the accuracy of the climatic data. We used daily climatic data based on the NCEP/NCAR Reanalysis 1 dataset and the NOAA Climate

Prediction Center Global Unified Gauge-Based Analysis dataset, which are of great help in improving the accuracy of forest fire prediction models.

*4.2. Comparative Analysis of the Relative Impact of Different Variables on the Occurrence of Forest Fires in Different Countries in the Cross-border Area between China, North Korea and Russia*

In general, the occurrence of forest fires in the cross-border area between China, North Korea and Russia is mainly affected by climatic factors, followed by topographic and vegetation factors, and finally by human activities. Previous studies have suggested that natural factors are usually considered to be the main factors affecting the occurrence of forest fires in large-scale studies [53]. Human activities may become a main factor behind forest fires in some other countries such as in Brazil, Indonesia, Thailand and other Southeast Asian countries. As the scale decreases, the degree of influence of human activities becomes more prominent [26]. This is because a large-scale study area is mainly dominated by the natural environment, and humans rarely set foot in many areas. What affects the occurrence of forest fires is the disaster-pregnant environment dominated by natural factors. However, in small- and medium-scale studies, the proportion of the human transformation of the natural environment and human activity areas is higher; therefore, more human factors need to be considered. From the perspective of these different countries, the impact of climatic factors on forest fires is higher on the Russian side than on the North Korean and Chinese sides. This is not only because the forest area that has not been affected by human activities on the Russian side is large and the population is small, but also because of Russia's laissez-faire forest fire management policy. This means that humans have a poor ability to prevent and mitigate forest fires on the Russian side [36], resulting in the occurrence, spread and extinguishing of forest fires that are more affected by climatic factors. The impact of human activity factors on forest fires is higher on the Chinese side than on the North Korean and Russian sides. This may be due to a higher human accessibility on the Chinese side due to the higher population density in China.

4.2.1. Relative Impact of Climatic Factors

From the perspective of the impact of climatic factors on the occurrence of forest fires, the daily cumulative precipitation is the most important factor affecting the occurrence of forest fires in the cross-border area between China, North Korea and Russia. Precipitation will directly affect the moisture content of combustibles, thereby reducing their combustibility. The relative impact of precipitation varies significantly within different countries. From the perspective of natural conditions, this may be because the Russian side is controlled by a temperate oceanic monsoon climate, while the Chinese and North Korean sides have a temperate continental monsoon climate. The eastern water vapor from the Sea of Japan is blocked by the mountains and mostly forms precipitation on the Russian side, resulting in much higher precipitation compared with the North Korean and Chinese sides. From the perspective of a forest fire management policy, due to the relatively passive forest fire management in Russia [36], human intervention in forest fires is relatively small, and forest fires are mostly extinguished by natural precipitation, which leads to the influence of precipitation on the Russian side being much higher than that on the North Korean side or the Chinese side.

The influence of the daily average temperature, daily average relative humidity and daily average wind speed on the occurrence of forest fires in the different countries is similar. A large number of studies have suggested that a high temperature will increase the temperature of combustibles; thereby, reducing the energy required for those combustibles to reach the ignition point, resulting in an increase in the probability of forest fires [19]. Relative humidity represents the amount of water vapor in the air, and lower water vapor levels in the air increase the likelihood of a forest fire burning [54]. Our findings are consistent with previous studies, but the relative effect of the daily mean wind speed was small in this study, and was not even selected as a significant variable on the Chinese side. This may be due to the fact that the wind speed mainly affects the intensity of a forest fire

and the fire area by increasing the propagation speed of the forest fire [55]; however, it has less influence on the occurrence of a forest fire.

### 4.2.2. Relative Impact of Topography and Vegetation Factors

From the perspective of the impact of topography factors on the occurrence of forest fires, the relative influence of the elevation is much greater than that of the slope. In previous studies, fires have been found to be more likely to occur in low-altitude areas [17], but the positive and negative correlations between the slope and forest fires were different in different studies [15,19]. To a certain extent, elevation affects the composition and spatial distribution of tree species (i.e., combustibles) by affecting the local air temperature and relative humidity, and indirectly affecting the occurrence of forest fires [56]. Higher altitudes, on the one hand, lead to lower air temperatures and a higher relative humidity. On the other hand, a high altitude reduces human accessibility, and the incidence of forest fires will also decrease. In this study, the influence of the elevation of the North Korean side was lower than that of the Chinese side and the Russian side. This may be due to the generally higher altitude on the North Korean side, which is above the Kaema Highlands as a whole. A large part of the food production for the local residents is from farming along the hillsides [57]; therefore, on the North Korean side, human activities in the high-altitude areas are no less than those in the low-altitude areas, and the high-altitude areas only suppress forest fires via their local climate. In general, the slope affects the retention time of precipitation, resulting in an uneven distribution of the fuel moisture content [58]. Furthermore, due to the flame attachment characteristics of forest fires, the steeper the slope, the faster the fire will spread and the higher the burning intensity will be [59]; however, in this study, the effect of the slope on forest fires was small, and it was only selected as a significant variable on the Chinese side. This may be due to the fact that, on the North Korean side and the Russian side, the terrain is not significantly undulating; therefore, the effect of the difference in the slope on forest fires is not obvious.

In this study, the FVC was calculated by the annual maximum EVI value, which represented the optimal vegetation coverage in a year, while the monthly EVI was biased to characterize the vegetation growth in different growth stages. Some studies have suggested that the optimal vegetation coverage largely determines the spatial distribution of fuel loads, because areas with a high vegetation coverage tend to accumulate more litter [60]; however, from these results, the impact of the annual FVC on the forest fire occurrence was much lower than that of the monthly EVI. This may be due to the fact that fires caused by poor vegetation growth are more frequent than fires caused by highly combustible loads in the China–North Korea–Russia border area. This can be demonstrated by the phenomenon of forest fires mostly occurring in spring and autumn [35]; as the moisture content of the vegetation itself is lower at the stage when the vegetation has not yet sprouted or withered, wildfires are more likely to burn and spread during this time.

### 4.2.3. Relative Impact of Human Activity Factors

The impact of human activities on the occurrence of forest fires is quite different in the different countries. On the Chinese side, the distance from a road and distance from a railway were not selected as significant variables. This is probably due to the fact that China actively promotes the importance of forest fire prevention and combats human arson, resulting in few fires around roads, and the airtight nature of today's trains also prevents fires caused by cigarette butts being thrown out during a train's journey. China also uses chemical agents with fire-fighting aircraft to put out fires [37], and is less dependent on roads and railways. The distance from buildings was negatively correlated with the forest fire occurrence, and the population density was positively correlated with the forest fire occurrence, indicating that fires mainly occurred near villagers' houses in high population areas. From the perspective of fire sources, the Chinese side has a large population that is concentrated in the low-altitude grain-growing areas. Although the State Environmental Protection Administration of China issued the "Administrative Measures

for the Prohibition of Burning and Comprehensive Utilization of Straw" in 1999, and the burning of straw has been prohibited in the delineated no-burning areas [61], agricultural fires still lead to some forest fires [62]; therefore, on the Chinese side, the closer to the densely populated area, the higher the probability of forest fires.

On the North Korean side, the distance from a railway was not selected as a significant variable, the distance from a road and population density were negatively correlated with the forest fire occurrence, and the distance from buildings was positively correlated with the forest fire occurrence. This may be due to the fact that people in North Korea mostly travel by foot or bicycle, and rarely take the train [63]. As a result, the highways have become a concentrated area of man-made fire sources, and railways have little impact on forest fires. In addition, due to the poor fire-fighting conditions in North Korea and the lack of modern fire-fighting equipment, when a large fire occurs, the government usually mobilizes nearby people to extinguish the fire, which requires a certain number of human resources [38]. Therefore, in areas with a high population density and close proximity to buildings, the probability of forest fires will decrease.

On the Russian side, population density was not selected as a significant variable, the distance from a road was positively correlated with a forest fire occurrence, and the distance from a railway and distance from buildings were negatively correlated with the forest fire occurrence. Due to the high latitude and thick permafrost in the Russian Far East, railway construction is difficult, and as such, there is only the Trans-Siberian Railway in the study area connecting Khabarovsk and Vladivostok. This area is lower in elevation and closer to grain-growing areas, making it more prone to forest fires. On the eastern side of Russia, the Sikhote-Alin Mountains have higher altitudes and lower temperatures, and the settlements are connected by roads, which makes the forest more fragmented. These roads act as a barrier to prevent the spread of forest fires [27].

*4.3. Spatial Distribution of High-Risk Areas for Forest Fire and Forest Fire Prevention and Control*

It can be seen from the fire risk level zoning map of the China–North Korea–Russia cross-border area that (Figure 4b), on the Chinese side, the high-risk forest fire areas are concentrated in the Sanjiang Plain in the northeast, and are scattered around the non-forest areas in the south. This is mainly due to the fact that most of the forest fires in China are caused by agricultural fires [62]. As the largest commercial grain production base in Northeast China, the Sanjiang Plain requires significant straw burning activities before spring ploughing every year, which greatly increases the risk of forest fires. Since the Sanjiang Plain was reclaimed as agricultural land, a large amount of the forest resources has been lost. If the remaining forests are not protected, this will cause incalculable losses to the natural ecology of the area. Since the promulgation of the ban on burning in Heilongjiang Province in 2020, the Sanjiang Plain has implemented a "comprehensive ban on burning in all regions and at all times", and resolutely prohibits any form of burning straw [64], which has a great effect on reducing man-made fire sources. At the same time, forest fire-warning weather stations and protective forest belts, dominated by fire-resistant tree species, should be established in high-risk areas to facilitate a green belt while preventing the large-scale spread of forest fires, causing great economic losses.

From the North Korean side, the high-risk forest fire areas are concentrated in Hamgyong-bukto Province in the northeast. On the Russian side, the high-risk forest fire area runs southward along the Ussuri River, the Sino–Russian border river, to the Muling–Xingkai Plain. For China, transboundary fire spread from North Korea and Russia also needs to be prevented and controlled. In particular, a forest fire in the Northeast Tiger Leopard National Park, located on the border between China, North Korea and Russia, would cause great damage to the forest resources and rare animals in the reserve. It is suggested that a high-resolution digital remote monitoring system be set up near the borderline to monitor the fire situation in the border area in real time and provide a timely warning. It is also recommended that inter-country forest firefighting cross-border rescue teams be formed, relying on the "Belt and Road" cooperation mechanism, to actively carry

out international emergency rescue cooperation and exchange. Additionally, this would help with communications and discussions between the neighboring countries to reach an agreement that firefighting aircraft can be sent across a border to fight fires when necessary. China, North Korea and Russia should actively strengthen their cooperation in the management and prevention of transboundary forest fires, and establish a joint and cooperative defense mechanism for such forest fires, to reduce the subsequent potential natural resource and economic losses.

## 5. Conclusions

In this study, based on the LR model, a prediction model for the probability of forest fire occurrence in different countries in the China–North Korea–Russia cross-border area was established. The relative influences of the driving factors of forest fire occurrence in this area were compared via the partially-standardized logistic regression coefficient, while the Kriging interpolation method was used to establish the forest fire occurrence probability and fire risk level division in the cross-border area between China, North Korea and Russia.

The results show that the LR model can accurately predict the probability of a forest fire in the cross-border area between China, North Korea and Russia. The model's goodness-of-fit AUC values were between 0.8973 and 0.9153, and the model prediction accuracy was between 81.23% and 85.08%. From the whole study area, the occurrence of forest fires in the cross-border areas of China, North Korea and Russia is mainly influenced by climatic factors, followed by topographic and vegetation factors, and finally, by human activities. From a country-by-country perspective, the forest fires on the Chinese side have been more anthropogenic than on the North Korean and Russian sides, and were mainly concentrated in low-elevation areas with high population densities. The forest fires on the North Korean side and the Russian side were more naturally affected than on the Chinese side, mainly occurring in areas with a low altitude, high temperature and little rainfall. The high-risk areas for forest fires were mostly concentrated near the borderline between China, North Korea and Russia. Due to the complexity of the geopolitics in the cross-border area between China, North Korea and Russia, transboundary firefighting has certain difficulties. In the future, it will be necessary to strengthen the cooperation between the countries, and to establish a joint prevention and cooperation mechanism for transboundary forest fires to protect the declining forest resources and the habitats of rare animals.

**Author Contributions:** Conceptualization, R.J. and D.Q.; methodology, R.J. and D.Q.; software, D.Q.; validation, R.J., D.Q. and H.Q.; formal analysis, D.Q.; investigation, D.Q.; resources, R.J.; data curation, D.Q.; writing—original draft preparation, D.Q.; writing—review and editing, R.J., H.Q., W.Z. and Z.L.; visualization, D.Q.; supervision, R.J.; project administration, R.J. and W.Z.; funding acquisition, R.J. and W.Z. All authors have read and agreed to the published version of the manuscript.

**Funding:** This research was funded by National Natural Science Foundation of China (41830643, 41807508, 42067065), and Jilin Provincial Science and Technology Department Project (20210101106JC, 20200403030SF, 20190201308JC).

**Conflicts of Interest:** The authors declare no conflict of interest.

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
