# Peer review of "A Comparative Study on the Drivers of Forest Fires in Different Countries in the Cross-Border Area between China, North Korea and Russia"

_forests, doi:10.3390/f13111939_

Round 1

Reviewer 1 Report

The authors presented a very interesting manuscript. However, I have a few comments.

1. The following publications should be included in the introduction: https://dx.doi.org/10.24189/ncr.2020.015; https://dx.doi.org/10.1038/s41598-021-00816-3; https://dx.doi.org/10.24189/ncr.2021.022; https://dx.doi.org/10.24189/ncr.2021.035

2. Line 127-128. What is the reason for the increase in the number of fires in Russia compared to China and North Korea?

3. Line 588-592. What are your suggestions on mechanisms for detecting and extinguishing fires in a transboundary zone?

Author Response

Dear reviewers:

We are very grateful to your comments for the manuscript. According with your advice, we tried our best to amend the relevant part and made some changes in the manuscript. These changes will not influence the content and framework of the paper. All of your questions were answered below. In the attachment, we list the modifications. And the revision marks were retained in the revised paper.

We appreciate for Reviewers’ warm work earnestly, and hope that the correction will meet with approval. Should you have any questions, please contact us without hesitate. 

Once again, thank you very much for your comments and suggestions.

Yours Sincerely,

Donghe Quan

Reviewer 2 Report

All comments and suggestions you can find in attached file 

Author Response

(The authors gave the same response as above.)

Reviewer 3 Report

Congratulation for this interesting paper. However, some clarifications and adding some items are needed to improve the quality of your paper, as shown in the attached file of result of my review. Good Luck. 

Author Response

(The authors gave the same response as above.)
